# Performance Baseline of Phase Transfer Entropy Methods for Detecting Animal Brain Area Interactions

**DOI:** 10.3390/e25070994

**Published:** 2023-06-29

**Authors:** Jun-Yao Zhu, Meng-Meng Li, Zhi-Heng Zhang, Gang Liu, Hong Wan

**Affiliations:** 1School of Electrical and Information Engineering, Zhengzhou University, Zhengzhou 450001, China; zhujunyao000@163.com (J.-Y.Z.); limengmeng1014@163.com (M.-M.L.); zhangzhiheng_zzu@163.com (Z.-H.Z.); 2Henan Key Laboratory of Brain Science and Brain-Computer Interface Technology, Zhengzhou 450001, China

**Keywords:** phase transfer entropy, interaction, visual–spatial associative learning, hippocampus, nidopallium caudolaterale, pigeon

## Abstract

***Objective:*** Phase transfer entropy ($TE^{\theta }$) methods perform well in animal sensory–spatial associative learning. However, their advantages and disadvantages remain unclear, constraining their usage. ***Method:*** This paper proposes the performance baseline of the $TE^{\theta }$ methods. Specifically, four $TE^{\theta }$ methods are applied to the simulated signals generated by a neural mass model and the actual neural data from ferrets with known interaction properties to investigate the accuracy, stability, and computational complexity of the $TE^{\theta }$ methods in identifying the directional coupling. Then, the most suitable method is selected based on the performance baseline and used on the local field potential recorded from pigeons to detect the interaction between the hippocampus (Hp) and nidopallium caudolaterale (NCL) in visual–spatial associative learning. ***Results:*** (1) This paper obtains a performance baseline table that contains the most suitable method for different scenarios. (2) The $TE^{\theta }$ method identifies an information flow preferentially from Hp to NCL of pigeons at the θ band (4–12 Hz) in visual–spatial associative learning. ***Significance:*** These outcomes provide a reference for the $TE^{\theta }$ methods in detecting the interactions between brain areas.

## 1. Introduction

Sensory–spatial associative learning is defined as the ability to integrate sensory cues (visual, olfactory, and auditory cues) and spatial locations together in memory [1,2,3]. Accumulating evidence shows that a distributed brain network supports animals in realizing this complex cognitive process [1,4,5], especially for the interaction between the hippocampus (Hp) and prefrontal cortex (PFC) [6,7], playing a key role in sensory–spatial associative learning. Avian Hp is homologous with mammalian Hp and the pigeons with Hp damage are impaired in forming an association between visual cues and spatial location [8,9], which indicates that the formation of associative memory critically depends on the integrity of Hp. As an analogue of mammalian PFC, pigeon nidopallium caudolaterale (NCL) integrates mnemonic information and task rules in order to direct behavior appropriately [10,11]. However, it is not yet understood whether there is information transfer between Hp and NCL of pigeons in sensory–spatial associative learning. Revealing the coupling between Hp and NCL not only helps to understand the mechanisms of brain areas, but also promotes the development of brain-like intelligence [12].

The interaction between Hp and NCL can be identified by phase transfer entropy ($TE^{\theta }$), which relates to calculating transfer entropy (TE) on the neural signals recorded from these two areas. $TE^{\theta }$ retains the advantages of TE because it is model-free and does not require any prior knowledge on the input of the system or the target connectivity networks [13,14,15]. Meanwhile, owing to the phase relationships linked to neuronal synchronization and information flow within the interconnected brain regions [16,17,18], compared with TE on the time series that combined amplitude and phase, $TE^{\theta }$ is more suitable for investigating coupling between brain regions.

In $TE^{\theta }$, the most critical assignment is to estimate the $TE^{\theta }$ metric value. According to its standard definition, $TE^{\theta }$ is formulated as Shannon entropy, which quantifies the difference between $p\left(\theta_{i+1}^{y}|\theta_{i}^{x},\theta_{i}^{y}\right)$ and $p\left(\theta_{i+1}^{y}|\theta_{i}^{x}\right)$ [15,19]. Hlaváčková-Schindler focused on diverse approaches to Shannon entropy [20]. Among them, a method is proposed by Kraskov, Stögbauer, and Grassberger (KSG) [21], in which the TE metric value can be estimated by counting the samples in specific strips, rather than finding nearest neighbors in each low-dimensional space [22], so that the deviations caused by the different spatial scales in low-dimensional spaces are significantly reduced. Another widely used approach to estimate $TE^{\theta }$ is the binning method [19]. By partitioning the state spaces of the phase time series into several bins and counting the number of points in each bin, various probability density functions (PDFs) and Shannon entropy can be calculated. Besides Shannon entropy, other techniques have also been proposed to estimate the $TE^{\theta }$ metric value; for instance, the symbolic technique based on permutation entropy and the kernel method with the concept of Renyi’s α entropy. Staniek suggested to estimate $TE^{\theta }$ by adopting a technique of symbolization [23]. Instead of calculating PDFs directly in traditional $TE^{\theta }$, a symbolic processing is carried out on the state space of phase time series [24]. With the relative frequencies of the symbols, the PDFs of the sequence of permutation index are estimated and the symbolic $TE^{\theta }$ metric value is obtained. Panche proposed a kernel $TE^{\theta }$ estimator that expresses $TE^{\theta }$ as a linear combination of Renyi’s entropy [25], which is approximated by a function defined on positive definite and infinitely divisible kernel matrices. This method sidesteps the requirement of obtaining the PDFs from the phase time series [26]. However, these methods mentioned above have their own assumptions, advantages, and limitations [25,26,27], and the criteria for applying these methods are different.

Owing to performance baseline differences in $TE^{\theta }$ methods, these methods may yield different results even for the same datasets, let alone the neural signals recorded during associative learning, where the short duration of a trial results in the small sample size and the interaction lag is unknown. Therefore, benchmarking $TE^{\theta }$ methods on various types of data to determine their performance baseline is an urgent problem to be solved. Though the ability of the $TE^{\theta }$ with KSG ($TE_{KSG}^{\theta }$), symbolic ($TE_{sym}^{\theta }$), and Renyi’s entropy ($TE_{k\alpha }^{\theta }$) estimators to detect the directional interaction has been discussed in [25], there is no uniform standard to evaluate the performance of these methods, and some key factors that may have a great effect on the estimators are not analyzed, especially for the sample size and interaction lag. The properties of the $TE^{\theta }$ methods for suitable application are not yet well understood.

Therefore, this paper first explores the performance baseline of $TE_{KSG}^{\theta }$, $TE_{sym}^{\theta }$, $TE_{k\alpha }^{\theta }$, and the binning method ($TE_{bin}^{\theta }$) on the simulated signals generated by a neural mass model and the actual neural data from ferrets with known interaction structures. To satisfy the validity for a non-stationary time series, we adopt the ensemble method in which multiple independent repetition trials are taken as a set to estimate $TE^{\theta }$, instead of estimating from an individual trial [28,29]. On the simulated data, the accuracy of the $TE^{\theta }$ methods in identifying the directional interaction and the ability of these methods to estimate the interaction lag is analyzed. The effect of sample size and interaction lag on these methods as well as the robustness of the $TE^{\theta }$ estimators to noise and linear mixing is also emphasized. Finally, the computational complexity of the four methods is compared. On the actual neural data, the $TE^{\theta }$ methods are applied to public datasets to assess the performance of $TE^{\theta }$ estimators on the actual neural signals.

Then, based on the properties, the most suitable $TE^{\theta }$ method is applied to the local field potential signals (LFPs) recorded from pigeons to detect the interaction between Hp and NCL in visual–spatial associative learning. The contributions of this paper are as follows: 

(1)By benchmarking the $TE^{\theta }$ methods on the simulated signals and the actual neural data, we explore the performance baseline of the $TE^{\theta }$ methods and provide a reference for the use of the $TE^{\theta }$ methods on the actual neural signals.(2)The most suitable $TE^{\theta }$ method is applied to neural signals recorded from pigeons and identifies the interaction between Hp and NCL of pigeons in visual–spatial associative learning.

The remainder of this paper is organized as follows: Section 2 reviews the theoretical foundations of the $TE^{\theta }$ methods, the neural mass model, evaluation criteria, and the actual neural data recording as well as analysis; Section 3 presents our results on the simulated signal pairs, actual neural data from ferrets, and the LFPs from pigeons; Section 4 shows the discussion; and, finally, Section 5 contains our conclusion.

## 2. Materials and Methods

### 2.1. Phase Transfer Entropy ($TE^{\theta }$)

For two random variables X and Y, the TE from X to Y can be defined as [30]:(1)TE(X→Y)=∑yt,yt−1dy,xt−udxp(yt,yt−1dy,xt−udx)logp(yt|yt−1dy,xt−udx)p(yt|yt−1dy)
where xt−udx, yt−1dy∈ℝD×d are the time-embedded series of X and Y, respectively, and D=T−(τ(d−1))−u, in which d,τ ∈ N are the embedding dimension and embedding delay, respectively. T indicates the length of X and u ∈ N represents the interaction lag between X and Y. p(·) is the PDF. Details of TE are described in [13,14,31].

In $TE^{\theta }$, the time series $x_{t}$ and $y_{t}$ are replaced by instantaneous phase time series $\theta_{t}^{x}$ and $\theta_{t}^{y}$, which are extracted from $x_{t}$ and $y_{t}$ with Hilbert or Morlet wavelet transform at frequency f. Thus, $TE^{\theta }$ is
(2)$$TE^{\theta }\left(X\to Y,\;f\right)=\sum_{\theta_{t}^{y},\theta_{t-1}^{y,dy},\theta_{t-u}^{x,dx}}p\left(\theta_{t}^{y},\theta_{t-1}^{y,dy},\theta_{t-u}^{x,dx}\right)\log\frac{p\left(\theta_{t}^{y}|\theta_{t-1}^{y,dy},\theta_{t-u}^{x,dx}\right)}{p\left(\theta_{t}^{y}|\theta_{t-1}^{y,dy}\right)}$$
where $\theta_{t-u}^{x,dx}$ and $\theta_{t-1}^{y,dy}$ are the time-embedded versions of $\theta_{t}^{x}$ and $\theta_{t}^{y}$.

Then, $TE^{\theta }$ is extended to the ensemble method, in which the independent repetition trials of an experimental condition are taken as an ensemble of realizations, and various PDFs are estimated from the ensemble members. Therefore, $TE^{\theta }$ can be written as follows [28]:(3)$$TE^{\theta }\left(X\to Y,f\right)=\sum_{\theta_{t}^{y}\left(r\right),\theta_{t-1}^{y,dy}\left(r\right),\theta_{t-u}^{x,dx}\left(r\right)}p\left(\theta_{t}^{y}\left(r\right),\theta_{t-1}^{y,dy}\left(r\right),\theta_{t-u}^{x,dx}\left(r\right)\right)\log\frac{p\left(\theta_{t}^{y}\left(r\right)|\theta_{t-1}^{y,dy}\left(r\right),\theta_{t-u}^{x,dx}\left(r\right)\right)}{p\left(\theta_{t}^{y}\left(r\right)|\theta_{t-1}^{y,dy}\left(r\right)\right)}$$
where r is the number of independent repetition trials.

### 2.2. $TE^{\theta }$ Methods

#### 2.2.1. Binning Estimator ($TE_{bin}^{\theta }$)

Formula (3) can be written as the form of Shannon entropy:(4)$$TE^{\theta }\left(X\to Y,\;f\right)=H\left(\theta_{t-u}^{x,dx}\left(r\right),\theta_{t-1}^{y,dy}\left(r\right)\right)-H\left(\theta_{t}^{y}\left(r\right),\theta_{t-u}^{x,dx}\left(r\right),\theta_{t-1}^{y,dy}\left(r\right)\right)+H\left(\theta_{t}^{y}\left(r\right),\theta_{t-1}^{y,dy}\left(r\right)\right)-H\left(\theta_{t-1}^{y,dy}\left(r\right)\right)\;$$
(5)$$H\left(\theta_{t-u}^{x,dx}\left(r\right),\theta_{t-1}^{y,dy}\left(r\right)\right)=-\sum_{\theta_{t-u}^{x,dx}\left(r\right),\theta_{t-1}^{y,dy}\left(r\right)}p\left(\theta_{t-u}^{x,dx}\left(r\right),\theta_{t-1}^{y,dy}\left(r\right)\right)\log\left(p\left(\theta_{t-u}^{x,dx}\left(r\right),\theta_{t-1}^{y,dy}\left(r\right)\right)\right)\;$$
(6)$$H\left(\theta_{t}^{y}\left(r\right),\theta_{t-u}^{x,dx}\left(r\right),\theta_{t-1}^{y,dy}\left(r\right)\right)=-\sum_{\theta_{t}^{y}\left(r\right),\theta_{t-u}^{x,dx}\left(r\right),\theta_{t-1}^{y,dy}\left(r\right)}p\left(\theta_{t}^{y}\left(r\right),\theta_{t-u}^{x,dx}\left(r\right),\theta_{t-1}^{y,dy}\left(r\right)\right)\log\left(p\left(\theta_{t}^{y}\left(r\right),\theta_{t-u}^{x,dx}\left(r\right),\theta_{t-1}^{y,dy}\left(r\right)\right)\right)\;$$
(7)$$H\left(\theta_{t}^{y}\left(r\right),\theta_{t-1}^{y,dy}\left(r\right)\right)=-\sum_{\theta_{t}^{y}\left(r\right),\theta_{t-1}^{y,dy}\left(r\right)}p\left(\theta_{t}^{y}\left(r\right),\theta_{t-1}^{y,dy}\left(r\right)\right)\log\left(\theta_{t}^{y}\left(r\right),\theta_{t-1}^{y,dy}\left(r\right)\right)$$
(8)$$H\left(\theta_{t-1}^{y,dy}\left(r\right)\right)=-\sum_{\theta_{t-1}^{y,dy}\left(r\right)}p\left(\theta_{t-1}^{y,dy}\left(r\right)\right)\log\left(\theta_{t-1}^{y,dy}\left(r\right)\right)$$
where H(·) is Shannon entropy.

The histogram-based method is used to estimate various PDFs in $TE_{bin}^{\theta }$. First, the state spaces of all trials ($\left(\theta_{t-u}^{x,dx}\left(r\right),\theta_{t-1}^{y,dy}\left(r\right)\right)$, $\left(\theta_{t}^{y}\left(r\right),\theta_{t-u}^{x,dx}\left(r\right),\theta_{t-1}^{y,dy}\left(r\right)\right)$, $\left(\theta_{t}^{y}\left(r\right),\theta_{t-1}^{y,dy}\left(r\right)\right)$, and $\left(\theta_{t-1}^{y,dy}\left(r\right)\right)$) can be divided into several bins. Then, the number of points in each bin is counted. Finally, the probability value for each bin is computed by dividing the number of points in that bin by the total number of data points.

The bin width is the only parameter should be determined in the binning method. According to Scott’s choice [32], bin width can be defined as follows: $h=3.5\sigma /N^{\frac{1}{3}}$, where N is the number of samples for $\theta_{t}^{y}$ and σ is the standard deviation for a directional variable as defined by Fisher. In $TE_{bin}^{\theta }$, the embedding dimension (dx*,*
dy) is 1.

#### 2.2.2. KSG Method ($TE_{KSG}^{\theta }$)

In the KSG estimator, the $TE^{\theta }$ metric value can be estimated by counting the number of samples in the strip of low-dimensional spaces. The strip is defined by the *k*th nearest neighbors in high-dimensional space projecting to the low-dimensional spaces. $TE_{KSG}^{\theta }$ can be written as follows:(9)$$TE_{KSG}^{\theta }\left(X\to Y,\;f\right)=\psi \left(k\right)+\langle \psi \left(n_{\theta_{t-1}^{y,dy}\left(r\right)}+1\right)-\psi \left(n_{\theta_{t}^{y}\left(r\right)\theta_{t-1}^{y,dy}\left(r\right)}+1\right)-\psi (n_{\theta_{t-1}^{y,dy}\left(r\right)\theta_{t-u}^{x,dx}\left(r\right)}+1)\rangle$$
where ψ is the Digamma function, $\psi \left(x\right)=\Gamma {\left(x\right)}^{-1}\frac{d\Gamma \left(x\right)}{dx}$. 〈·〉 means average. $n_{\theta_{t-1}^{y,dy}\left(r\right)}$, $n_{\theta_{t}^{y}\left(r\right)\theta_{t-1}^{y,dy}\left(r\right)}$, and $n_{\theta_{t-1}^{y,dy}\left(r\right)\theta_{t-u}^{x,dx}\left(r\right)}$ are the number of samples falling into the strip of low-dimensional space $\theta_{t-1}^{y,dy}\left(r\right)$, $\theta_{t}^{y}\left(r\right)\theta_{t-1}^{y,dy}\left(r\right)$, and $\theta_{t-1}^{y,dy}\left(r\right)\theta_{t-u}^{x,dx}\left(r\right)$, respectively. k is generally 4 [21].

In the KSG method, the Rawdgitz criterion is used to calculate the embedding dimension and embedding delay [33].

#### 2.2.3. Symbolic Estimator ($TE_{sym}^{\theta }$)

The symbolic version of $TE^{\theta }$ is based on permutation entropy [34]. A coarse-graining (partitioning) of state space method is used, in which the phase spaces of $\theta_{t}^{x}$ and $\theta_{t}^{y}$ are arranged in an ascending order and uniquely mapped onto one of the possible permutations. Then, the symbols are defined and, with the relative frequency of symbols, the joint and conditional probabilities of the sequence of permutation indices can be calculated. The symbolic $TE^{\theta }$ method is applied to the ensemble repetitions:(10)$$TE_{sym}^{\theta }\left(X\to Y,f\right)=\sum_{{\hat{\theta }}_{i}^{y}\left(r\right),{\hat{\theta }}_{i-1}^{y}\left(r\right),{\hat{\theta }}_{i-u}^{x}\left(r\right)}p\left({\hat{\theta }}_{i}^{y}\left(r\right),{\hat{\theta }}_{i-1}^{y}\left(r\right),{\hat{\theta }}_{i-u}^{x}\left(r\right)\right)\log\frac{p\left({\hat{\theta }}_{i}^{y}\left(r\right)|{\hat{\theta }}_{i-1}^{y}\left(r\right),{\hat{\theta }}_{i-u}^{x}\left(r\right)\right)}{p\left({\hat{\theta }}_{i}^{y}\left(r\right)|{\hat{\theta }}_{i-1}^{y}\left(r\right)\right)}$$
where ${\hat{\theta }}_{i}^{x}$ and ${\hat{\theta }}_{i}^{y}$ are the results of $\theta_{t}^{x}$ and $\theta_{t}^{x}$ symbolization.

In the process of phase space reconstruction, the C–C method is applied to calculate the embedding dimension and embedding delay [35].

#### 2.2.4. Renyi’s α–Entropy Estimator ($TE_{k\alpha }^{\theta }$)

For two-phase time series θtx and θty, TEθ from X to Y can be expressed as the kernel-based formulation of Renyi’s α–order entropy:(11)TEκαθ(X→Y,f)=Hα(Κθt−1y,dy(r),Κθt−ux,dx(r))−Hα(Κθty(r),Κθt−1y,dy(r),Κθt−ux,dx(r))−Hα(Κθty(r),Κθt−1y,dy(r))+Hα(Κθt−1y,dy(r))
where Hα(A,B)=Hα(A°Btr(A°B))=11−αlog(tr((A°Btr(A°B))α)) is Renyi’s α-order entropy. tr(·) stands for matrix trace. Κθty(r), Κθt−1y,dy(r), and Κθt−ux,dx(r) are the Gram matrixes for θty(r), θt−1y,dy(r), and  θt−ux,dx(r), respectively, and hold elements kij=κ(ai,aj), κ(ai,aj)=exp(−∥ai−aj∥22σ2). For Κθty(r), ai, aj∈ℝ are the values of θy at times i and j. In the case of matrix Κθt−1y,dy(r), the vectors ai, aj∈ℝd contain the space state reconstruction θy,dy of θy at times i and j, likewise for Κθt−ux,dx(r). ∥·∥ indicates Euclidean distance and σ is defined as the kernel bandwidth, which is calculated by the median of ∥ai−aj∥.

The Rawdgitz criterion is used to calculate the embedding dimension and embedding delay, and α = 3 is chosen for Renyi’s α entropy by experience.

For all estimators, $TE^{\theta }$ is a biased estimation, and it may be non-zero even in the absence of an interaction between X and Y. To reduce the bias, we define differential $TE^{\theta }$ ($dTE^{\theta }$) for all $TE^{\theta }$ types:(12)$$dTE^{\theta }\left(X\to Y,f\right)=TE^{\theta }\left(X\to Y,f\right)-TE^{\theta }\left(Y\to X,f\right)$$

$dTE^{\theta }\left(X\to Y,f\right)$ > 0 indicates information flows preferentially from X to Y and $dTE^{\theta }\left(X\to Y,f\right)$ < 0 indicates the reverse direction. In the case of no preferential direction of interaction, $dTE^{\theta }\left(X\to Y,f\right)$ = 0.

To test the statistical significance of the $dTE^{\theta }$ value, the source variable X is shuffled to generate the surrogate data, and the $dTE^{\theta }$ values for 200 sets of surrogate data are calculated to construct the null hypothesis distribution. The null hypothesis of the raw data can be rejected or retained by comparing the $dTE^{\theta }$ value of the raw data to the null hypothesis distribution at the 1% level of significance.

The interaction lag δ between X and Y is a significant parameter. Here, the scanning method is used to estimate δ. When $TE^{\theta }\left(X\to Y,f,u\right)$ is maximal [30], u is equal to δ (Equation (13)).
(13)$$\delta ={\mathrm{argmax}}_{u}\left(TE^{\theta }\left(X\to Y,f,u\right)\right)$$

The Matlab codes of the $TE^{\theta }$ methods are provided in the Appendix A.

### 2.3. Neural Mass Model (NMM)

Simulated data play an essential role in evaluating the competing methods against a “ground truth”. An NMM is used to generate the simulated signal pairs with known interaction properties [36]. The NMM simulates the connectivity between multiple regions of interest (ROIs) through long-range excitatory connections. In the NMM, the average spike density of pyramidal neurons of the presynaptic area ($Z^{X}$) affects the postsynaptic area by a weight factor ω and a time delay δ (Equation (14)):(14)$$v^{Y}\left(t\right)=\omega^{X\to Y}Z^{X}\left(t-\delta \right)+n^{Y}\left(t\right)$$
where the superscripts X and Y are represented by the source and target region, respectively. n(t) is a Gaussian white noise. 

Signal pairs generated by the NMM are nonlinear and have significant β (about 20–30 Hz) activity (Figure 1). By changing the interaction delay and weight factor $\omega^{X\to Y}$, $\omega^{Y\to X}$, the simulated signal pairs with directional interaction can be obtained. In the following analysis, the signal pairs are first filtered in a 15 to 35 Hz pass band using a finite impulse response filter (FIR) with order 15, and then Hilbert transform is used to extract the phase time series from the filtered signals.

To test the accuracy of the $TE^{\theta }$ methods in detecting the directional interaction for different levels of coupling strength, $\omega^{Y\to X}$ is set to 0 and $\omega^{X\to Y}$ from 0 to 70 to simulate unidirectional coupling. The interaction lag δ is set to 20 ms. Then, the stability of these methods in the presence of noise and linear mixing is investigated. We use the method in [19] to add noise and linear mixing. The signal-to-noise ratio (SNR) is set to 30, 20, 10, 5, 0, and −5 dB, respectively, and the mixing strength varies from 0.1 to 0.5. For each case, 1000 simulated signal pairs are generated with a duration of 2 s. To investigate the impact of sample size on the performance of the competing estimators, 25, 50, 75, 100, 125, 150, 175, 200, 250, and 300 signal pairs are selected to simulate varied sample size. Finally, the interaction lag is changed from 10 to 50 ms in five steps to assess the effect of interaction lag on the $TE^{\theta }$ methods.

### 2.4. Evaluation Criteria

The $dTE^{\theta }$ value is applied to measure the coupling strength, and the false positive rate (FPR) and sensitivity are calculated to assess the accuracy of the $TE^{\theta }$ estimators in identifying the direction. By comparing the $dTE^{\theta }$ value of the raw data to the null hypothesis distribution, which is constructed by 200 sets of surrogate data at the significance p (0.01) level, FPR for $\omega^{X\to Y}$ = 0 is obtained. The sensitivity (proportion of true positive for $\omega^{X\to Y}\neq 0$) is calculated as a function of $\omega^{X\to Y}$ and the coupling detection threshold (CDT) for a sensitivity of 0.8 is estimated by linear interpolation. The CDT value represents the smallest coupling value for which the estimators detect 80% of the directional interaction. Therefore, a low CDT value indicates that a significant interaction is detected even for weak coupling, while a high CDT value means that the coupling could be detected only for solid coupling. δ accuracy is used to indicate the accuracy of these methods for interaction lag.

### 2.5. The Actual Neural Signals from Ferrets

The LFPs in the PFC and primary visual cortex (V1) of a female ferret in an awake state are recorded by a single metal electrode that is accurately inserted into putative layer IV. Details of the experimental process are described in [37] and the raw data are available from http://dx.doi.org/10.5061/dryad.kk40s (accessed on 1 November 2022).

Several sessions are carried out and the LFPs from each session are separated into epochs with a length of 4.8 s. Before detecting the interactions between PFC and V1, data processing is performed to remove the epochs containing motion artifact, and the epochs with large power in the delta (0.5 to 4 Hz) band (higher than 30% of the total power in 0.5 to 50 Hz) are also rejected. Then, the LFPs are filtered in 0–20, 20–40, 40–60, 60–80, and 80–100 Hz using a two-way, zero-phase-lag FIR filter with the order defined as 3 r, where r is the ratio of the sampling rate to the low-frequency cutoff of the filter, rounded down [38]. The phase time series of each frequency band can be obtained by Hilbert transform. Finally, the $TE^{\theta }$ methods are applied to the phase time series for identifying the interaction between PFC and V1.

To explore the performance baseline of these methods on the actual neural signals with a small and large sample size, 2 and 30 epochs are drawn randomly from each session as a set to estimate $TE^{\theta }$, respectively, and search the interaction lag in the range of 1 to 20 ms. This procedure is repeated 10 times and results in 10 $dTE^{\theta }$ values for each session. We first test the $dTE^{\theta }$ values for their significance (p < 0.05) within individual sessions against 200 surrogate datasets at each frequency band, and then use a binomial test to establish the statistical significance over recordings. 

### 2.6. The LFP Recorded from Pigeons and Analysis

The LFPs are recorded from Hp and NCL of six pigeons while they perform a visual–spatial associative learning task in a Y maze (Figure 2a). Figure 2c shows the schematic of the spatial associative learning task. A detailed description of subjects, surgical implantation of electrodes, data acquisition, and behavioral tasks are provided in the Appendix A.

Data processing is performed before analysis. First, combined with video, only the trials in which pigeons are ready to enter the next trial in the inter-trial interval (ITI) are reserved. The trials containing strong motion artifact are also removed. Then, the adaptive common average reference is used for all channels of the remaining trials to remove the spatially correlated noise [39]. θ (4–12 Hz), β (12–30 Hz), slow-γ (30–45 Hz), and fast-γ (55–80 Hz) bands of LFPs are extracted by the zero-phase-lag FIR filter with the order defined as 3 r, where r is the ratio of the sampling rate (2000 Hz) to the low-frequency cutoff of the filter, rounded down. The phase time series for each frequency band is extracted by Hilbert transform. Finally, the phase time series from ITI (the last 2 s) to the turning period (a total of 7 s) is divided into 14 non-overlapping bins, and the coupling between Hp and NCL is calculated for each bin.

For each pigeon, only one session (about 30 trials) with a correct rate of 75–80% is applied to detect the interactions in brain areas (Table 1). All trials in a session are assumed to have the equivalent brain activity. Thus, LFPs (recorded with two 16-channel microelectrode arrays from Hp and NCL) for all of the correct trials in a session are pooled together. For each frequency band and each bin, 200 signal pairs are drawn randomly to estimate $TE^{\theta }$ and the interaction lag is searched in the range of 1 to 30 ms. This process is repeated 30 times. The significance of $dTE^{\theta }$ value is tested against those of 200 surrogate datasets (p < 0.05) for each estimation, and then the binomial test is used to establish the statistical significance over all estimations.

## 3. Results

### 3.1. The Performance Baseline Based on the Simulated Data

#### 3.1.1. Accuracy of the $TE^{\theta }$ Methods for the Directional Interaction Detection

To investigate the accuracy of the competing methods for the directional interaction recognition, the simulated signal pairs with $\omega^{X\to Y}$ = 0, 10, 20, 30, 40, 50, 60, and 70; $\omega^{Y\to X}$ = 0; δ = 20 ms; and trial length = 2 s are used. For each $\omega^{X\to Y}$ value, 1000 simulated signal pairs are pooled together, and 100 pairs are drawn randomly as a set to estimate $TE^{\theta }$ and search for the interaction lag δ in the range of 10 to 70 ms. The $dTE^{\theta }$ distribution, FPR, sensitivity, and δ accuracy of the $TE^{\theta }$ methods are computed by 200 sets of signal pairs. This procedure is repeated 20 times to result in the mean and variance of the FPR, CDT values, and δ accuracy.

As shown in Figure 3, the $dTE^{\theta }$ values are clustered around 0 for $\omega^{X\to Y}$ = 0 and increase monotonically with $\omega^{X\to Y}$ from 0 to 70 for all methods. The $dTE^{\theta }$ values for $TE_{KSG}^{\theta }$ in the range of 0 to 0.05 are lower than those of other methods. The FPR values of the $TE^{\theta }$ methods have no significant difference and are at a low level (below 0.01), but there is a dramatic difference in the sensitivity of these estimators. The CDT values of $TE_{bin}^{\theta }$ and $TE_{sym}^{\theta }$ are around 17.5, which are significantly lower than those of other methods. The second is $TE_{k\alpha }^{\theta }$, followed by $TE_{KSG}^{\theta }$. The CDT values of $TE_{KSG}^{\theta }$ are higher than 50, indicating that the validity of $TE_{KSG}^{\theta }$ in the directional coupling detection is lower than other methods. Finally, the δ accuracy of these methods is also analyzed. For $TE_{bin}^{\theta }$, $TE_{KSG}^{\theta }$, and $TE_{k\alpha }^{\theta }$, the δ accuracy is increased in $\omega^{X\to Y}$. $TE_{bin}^{\theta }$ and $TE_{k\alpha }^{\theta }$ have the same performance in detecting the interaction lag, followed by $TE_{KSG}^{\theta }$. The δ values estimated by the $TE_{sym}^{\theta }$ method are clustered around 30 ms rather than 20 ms, as we set. Thus, $TE_{sym}^{\theta }$ cannot estimate the interaction lag correctly.

To sum up, $TE_{bin}^{\theta }$, $TE_{sym}^{\theta }$, and $TE_{k\alpha }^{\theta }$ can accurately detect the directional interaction with a low FPR and high sensitivity, but $TE_{sym}^{\theta }$ cannot identify the interaction lag correctly. Though $TE_{KSG}^{\theta }$ behaves well for $\omega^{X\to Y}$ = 0, it has poor reliability in the directional identification for $\omega^{X\to Y}$ > 0.

#### 3.1.2. Stability of the $TE^{\theta }$ Methods to the Directional Interaction Estimation

(1)Robustness of the $TE^{\theta }$ methods to the directional interaction identification in the presence of noise and linear mixing.

Noise is inevitably introduced into the process of neural signals’ acquisition and, owing to volume conduction between adjacent brain regions, there may be linear mixing in the neural signals. To investigate whether the competing $TE^{\theta }$ methods can reliably detect the strength and the direction of the interaction in the presence of noise and linear mixing, the simulated signal pairs ($\omega^{X\to Y}$ = 0, 10, 20, 30, 40, 50, 60, and 70; $\omega^{Y\to X}$ = 0; δ = 20 ms; trial length = 2 s) with varied SNR and linear mixing values (m) are applied for $TE^{\theta }$ estimation.

Figure 4 shows the behavior of estimators with SNR = 30, 20, 10, 5, 0, and −5 dB, with m = 0. Low power noise (SNR = 30, 20 dB) has a little effect on the $dTE^{\theta }$ values of the competing methods, and the $dTE^{\theta }$ values decrease moderately for SNR = 10, 5 dB. However, for strong noise (SNR = 0, −5 dB), they reduce greatly. The FPR values of the four estimators are not affected by noise. They all remain at a low level regardless of the noise power. However, the sensitivity of these estimators is seriously influenced by noise. Especially for $TE_{KSG}^{\theta }$ and $TE_{k\alpha }^{\theta }$, the direction cannot be detected even for a strong interaction when SNR is −5 dB. Although $TE_{bin}^{\theta }$, $TE_{sym}^{\theta }$, and $TE_{k\alpha }^{\theta }$ behave well for an SNR higher than 5 dB, their CDT values increase sharply when the SNR is lower than 5 dB. The δ accuracy for $TE_{bin}^{\theta }$, $TE_{KSG}^{\theta }$, and $TE_{k\alpha }^{\theta }$ shows the same trend as that of sensitivity, which increases with $\omega^{X\to Y}$ from 0 to 70 and reduces with an increase in noise. The δ accuracy of $TE_{bin}^{\theta }$ as well as $TE_{k\alpha }^{\theta }$ is significantly higher than that of other methods, followed by $TE_{KSG}^{\theta }$, and $TE_{sym}^{\theta }$ has the worst performance in identifying the interaction lag.

Figure 5 shows the effect of linear mixing (m = 0.1, 0.2, 0.3, and 0.5; SNR = 20 dB) on the $TE^{\theta }$ methods. The $dTE^{\theta }$ values of the estimators increase with $\omega^{X\to Y}$ from 0 to 70 even in the presence of linear mixing, and they are not affected by a low mixing strength (m = 0.1, 0.2, and 0.3), but are reduced for a strong mixing strength (m = 0.5). The FPR and the sensitivity of the competing methods with varied linear mixing strength are calculated. The FPR values of $TE_{k\alpha }^{\theta }$ remain at a low level regardless of the mixing strength. Those of $TE_{bin}^{\theta }$ and $TE_{KSG}^{\theta }$ are below 0.01 for m = 0.1 and 0.2 and are higher than 0.01 when m is 0.3 and 0.5. The FPR values of $TE_{sym}^{\theta }$ increase monotonically with mixing strength from 0.1 to 0.5 and reach 0.38 for m = 0.5. However, the sensitivity of $TE_{bin}^{\theta }$, $TE_{KSG}^{\theta }$, and $TE_{sym}^{\theta }$ shows the opposite trend and reduces with an increase in m. The CDT values of $TE_{bin}^{\theta }$ and $TE_{sym}^{\theta }$ are below 20 for m = 0.1 and 0.2 and increase to 30 when m is 0.5. Those of $TE_{KSG}^{\theta }$ are lower than 50 for m = 0.1, 0.2, and 0.3, but $TE_{KSG}^{\theta }$ cannot detect the directional interaction when m is 0.5. Linear mixing has little effect on $TE_{k\alpha }^{\theta }$ and the CDT values remain at a low level regardless of m. However, the δ accuracy of $TE_{bin}^{\theta }$ and $TE_{k\alpha }^{\theta }$ decreases with m from 0.1 to 0.5. That of $TE_{KSG}^{\theta }$ presents unexpected results and improves with an increase in m. $TE_{sym}^{\theta }$ cannot detect the interaction lag irrespective of the effect of linear mixing (m).

Above all, low, realistic noise (SNR = 30, 20, 10, and 5 dB) and mixing (m = 0.1 and 0.2) have little effect on the $TE^{\theta }$ methods, but the validity of the estimators is greatly reduced for strong noise and mixing. Compared with other methods, $TE_{bin}^{\theta }$ with a low false positive rate and high sensitivity performs the best for strong noise, while the robustness of $TE_{k\alpha }^{\theta }$ to linear mixing is better than that of other methods. The FPR of $TE_{sym}^{\theta }$ is seriously influenced by linear mixing. The performance of $TE_{KSG}^{\theta }$ is the most unstable and the directional interaction cannot be detected for strong noise as well as linear mixing.

(2)The impact of sample size on the performance of the $TE^{\theta }$ methods.

An accurate $TE^{\theta }$ estimation requires enough samples. We, therefore, investigate the effect of sample size on the performance of the $TE^{\theta }$ methods by drawing varied trials as a subset to estimate $TE^{\theta }$ (trial number = 25, 50, 75, 100, 125, 150, 175, 200, 250, and 300; trial length = 2 s). It should be emphasized that the Gram matrices’ operation in $TE_{k\alpha }^{\theta }$ requires a large amount of internal memory, so only small samples (trial number = 25, 50, 75, 100, and 125) are used in $TE_{k\alpha }^{\theta }$.

As expected, the mean values of $dTE^{\theta }$ are not affected by sample size and scale up monotonically with $\omega^{X\to Y}$ from 0 to 70, but the variances reduce with an increase in sample size, which indicates that a large sample size improves the stability of the $TE^{\theta }$ methods on the coupling strength quantification (Figure 6). The FPR values of $TE_{bin}^{\theta }$, $TE_{KSG}^{\theta }$, and $TE_{k\alpha }^{\theta }$ remain at a low level irrespective of the sample size. However, those of $TE_{sym}^{\theta }$ increase with the sample size from 5 × 10^3^ to 60 × 10^3^ and are higher than 0.05 when the sample size is 60 × 10^3^. A large sample capacity has a positive effect on the sensitivity of all methods and the CDT values of $TE_{bin}^{\theta }$, $TE_{sym}^{\theta }$, and $TE_{k\alpha }^{\theta }$ below 15 for sample size = 60 × 10^3^, indicating that these methods can detect the directional interaction even for weak coupling strength under the condition of enormous sample size. The δ accuracy of $TE_{bin}^{\theta }$, $TE_{KSG}^{\theta }$, and $TE_{k\alpha }^{\theta }$ is also improved by the sample size, but even for large samples, $TE_{sym}^{\theta }$ is still unable to correctly identify the interaction lag. 

(3)The effect of interaction lag on the $TE^{\theta }$ methods.

The interaction lag between brain areas is unknown. To analyze whether the validity of these estimators is affected by interaction lag, the simulated signal pairs ($\omega^{X\to Y}$ = 0, 10, 20, 30, 40, 50, 60, and 70; $\omega^{Y\to X}$ = 0; SNR = 20 dB; m = 0.1; trial length = 2 s) with varied interaction lag (δ = 10, 20, 30, 40, and 50 ms) are generated by the NMM. For each estimation, we draw 100 pairs as a set from 1000 signal pairs and scan the interaction lag in the range of 10 to 70 ms.

As shown in Figure 7, the $dTE^{\theta }$ values of these methods display different changing characteristics. Those of $TE_{bin}^{\theta }$ and $TE_{KSG}^{\theta }$ scale up with an increase in interaction lag. While for $TE_{sym}^{\theta }$ and $TE_{k\alpha }^{\theta }$, the $dTE^{\theta }$ values are at the maximum when the interaction lag is in the middle (δ = 30 ms) and decrease in both directions (δ = 10, 20, 40, and 50 ms). The FPR values of these methods are not influenced by interaction lag, and they are below 0.01. The sensitivity of $TE_{KSG}^{\theta }$ is improved for δ from 10 to 50 ms, and the CDT values decrease from higher than 70 to 25, indicating that $TE_{KSG}^{\theta }$ may be more suitable for the neural signals with large interaction lag. The sensitivity of $TE_{sym}^{\theta }$ is the best when δ is 30 ms and reduces on both sides. For $TE_{bin}^{\theta }$ and $TE_{k\alpha }^{\theta }$, the CDT values for δ = 10 ms are higher than those of other δ values. Finally, the δ accuracy of the estimators is calculated. The interaction lag has a great effect on the δ accuracy of $TE_{bin}^{\theta }$ and $TE_{KSG}^{\theta }$. $TE_{sym}^{\theta }$ cannot detect the interaction lag, irrespective of the analysis lag. 

#### 3.1.3. The Computational Complexity of the $TE^{\theta }$ Methods

$TE^{\theta }$ is a biased estimation and the $dTE^{\theta }$ values are not zero even in the absence of interaction, so it is necessary to construct the null hypothesis distribution to test the significance of the $dTE^{\theta }$ metric value. Generally, the surrogate signal pairs are shuffled 200 times to construct a null hypothesis distribution. So, the $TE^{\theta }$ methods should be time-saving. Here, the time consumption of these $TE^{\theta }$ estimators with varied sample sizes is compared. One complete directional interaction detection includes estimating the $dTE^{\theta }$ value as well as scanning the interaction lag in the range of 10 to 70 ms and constructing the null hypothesis distribution with 200 surrogate signal pairs.

The results are shown in Figure 8. The time consumption of the methods is scaled up with an increase in sample size. $TE_{bin}^{\theta }$ consumes the least time. It takes only 20.1118 s for sample size = 5 × 10^3^ and increases to 124.3577 when the sample capability is 60 × 10^3^. $TE_{sym}^{\theta }$ also has an acceptable time consumption (167.1969 s for sample size = 5 × 10^3^ and 524.7998 s for 60 × 10^3^). The third is $TE_{KSG}^{\theta }$ (194.9923 s and 5.2436 × 10^3^ s for sample capability = 5 × 10^3^ and 60 × 10^3^, respectively). $TE_{k\alpha }^{\theta }$ has the highest time cost. It takes 4.2735 × 10^4^ s for sample size = 5 × 10^3^, which is much greater than $TE_{bin}^{\theta }$. Thus, $TE_{k\alpha }^{\theta }$ may not be suitable for the signal pairs with a large sample size. 

### 3.2. The Performance Baseline of the $TE^{\theta }$ Methods Based on the Actual Neural Signals

To explore the properties of the $TE^{\theta }$ estimators on the actual neural signals, we apply the methods to the LFPs, which are recorded from PFC and V1 of a ferret in an awake state with a large and small sample size. Owing to the large time consumption of $TE_{KSG}^{\theta }$ and $TE_{k\alpha }^{\theta }$, only the small sample size is used on $TE_{KSG}^{\theta }$ and $TE_{k\alpha }^{\theta }$. 

For the large sample size (30 epochs as a set to estimate $TE^{\theta }$), $TE_{bin}^{\theta }$ as well as $TE_{sym}^{\theta }$ detect bidirectional coupling between PFC and V1 for the ferret in an awake state, and the information flows preferentially from V1 to PFC (known as feed-forward FF) in the low-frequency band (0–20 Hz), with p < 0.001 for $TE_{bin}^{\theta }$ and p < 0.01 for $TE_{sym}^{\theta }$, respectively, while the direction is reversed (from PFC to V1, called as feedback FB) in the γ band (40–60 Hz) with p < 0.001 (Figure 9). These results are in line with previous studies finding that there are bidirectional interactions between PFC and V1 in ferret brains in an awake state [40]. The δ values of $TE_{bin}^{\theta }$ for FF are clustered around 5 and 20 ms, and those of FB fluctuate around 1 ms (almost consistent with the results in [40]). However, the δ values of $TE_{sym}^{\theta }$ are distributed in the range of 0 to 20 ms, more dispersed than those of $TE_{bin}^{\theta }$. No method can identify the coupling between PFC and V1 with a small sample size (2 epochs as a set to estimate $TE^{\theta }$), except for $TE_{sym}^{\theta }$. An information flow predominantly from PFC to V1 at 60–80 Hz is detected by $TE_{sym}^{\theta }$. However, owing to its disappearance with an increase in sample size, it may be a false positive. The outcomes on the actual neural data are in line with those on the simulated signal pairs. Sample size has a positive effect on the applicability criteria of $TE^{\theta }$ methods, and $TE_{bin}^{\theta }$ performs better than other methods. $TE_{sym}^{\theta }$ may produce a high false positive rate, and the large time consumption of $TE_{k\alpha }^{\theta }$ hinders its application in neural signals.

Based on the above analysis, the performance baseline of these methods is proposed in Table 2.

### 3.3. Implementing a Suitable $TE^{\theta }$ Estimation Method on the Interaction between Hp and NCL of Pigeons in Visual–Spatial Associative Learning

$TE_{bin}^{\theta }$ is selected by the performance baseline table to explore the coupling between Hp and NCL of pigeons in spatial associative learning. The $TE_{bin}^{\theta }$ method is applied on the LFPs recorded from Hp and NCL of six pigeons (p21 to p26) while they perform a visual–spatial associative learning task. The LFPs from ITI to the turning period (a total of 7 s) are divided into 14 non-overlapping bins, then the $dTE^{\theta }$ value is calculated, and the null hypothesis distribution is constructed for each bin.

The results are shown in Figure 10. For all pigeons, the $dTE^{\theta }$ values in the period where the door is opened and animals enter the critical decision-making place are actually larger than those in other periods for the θ band (4–12 Hz). By comparing the $dTE^{\theta }$ value with the null hypothesis distribution for each bin, we find there exists coupling between Hp and NCL in the decision-making period at the θ band, and the information flow is predominately from Hp to NCL. The scan method is used to detect the interaction lag between Hp and NCL, finding that different pigeons have varied interaction lag values. For pigeons p21 to p26, the interaction lag values are clustered around 27, 23, 15, 24, 11, and 29 ms, respectively (Figure 10g). The coupling between Hp and NCL during associative learning may indicate that the spatially related associative information formed in Hp is transmitted to NCL for decision-making.

## 4. Discussion

This paper investigates the performance baseline of four commonly used $TE^{\theta }$ methods ($TE_{bin}^{\theta }$, $TE_{KSG}^{\theta }$, $TE_{sym}^{\theta }$, and $TE_{k\alpha }^{\theta }$) on the simulated signals and actual neural data. The results are shown in Table 2 and, for a large sample size, $TE_{bin}^{\theta }$ with less time consumption, a low false positive rate, and high sensitivity performs better than other methods. $TE_{k\alpha }^{\theta }$ is robust to linear mixing. It is a good choice for small samples. $TE_{sym}^{\theta }$ is extremely susceptible to linear mixing and cannot accurately detect the interaction lag. The performance of $TE_{KSG}^{\theta }$ is unstable and easily affected by interaction lag. Then, the $TE_{bin}^{\theta }$ method is applied to the LFPs recorded from Hp and NCL of pigeons to detect the coupling between these two brain regions while pigeons perform a visual–spatial associative learning task. $TE_{bin}^{\theta }$ identifies an information flow from Hp to NCL at the θ band when pigeons enter the critical decision-making place.

Recently, $TE^{\theta }$ has become a research focus in identifying animal brain area interactions, and various $TE^{\theta }$ methods have been proposed to estimate the $TE^{\theta }$ metric value. However, these methods with distinct underlying mathematical assumptions or measures of dependency show various characteristics. The characterization of $TE^{\theta }$ methods is unclear, which limits their use in the actual neural signals. This paper explores the performance baseline of $TE^{\theta }$ methods and provide a reference for researchers when using these methods, promoting the usage of $TE^{\theta }$ methods in neural signals. 

The accuracy of $TE^{\theta }$ estimators in identifying and quantifying the directional interaction is first analyzed in an ideal state (no noise and linear mixing). The estimators show almost the same results, except for $TE_{KSG}^{\theta }$, in which the validity is significantly lower than that of other methods for weak coupling. Transfer entropy with the KSG estimator ($TE_{KSG}$) is a widely used method for signal pairs combining amplitude and phase [31]. Though some researchers note it may be not applicable for phase time series because of the periodicity of the phase spectrum [20], the KSG estimator has been successfully used to estimate mutual information on phase time series [41]. In this study, $TE_{KSG}^{\theta }$ behaves well for a large interaction lag (δ = 50 ms). Therefore, $TE_{KSG}^{\theta }$ may be more suitable for analyzing the coupling with a large interaction lag.

The neural signals are corrupted by measurement noise and biological noise in their acquisition and transmission [39]. Though some measures have been taken to reduce the noise [42], we cannot take them out completely from neural signals owing to some noise being sufficiently complex. The presence of noise can mask the features of the neural signals and affect the analysis of the coupling between them. So, the $TE^{\theta }$ methods should be robust to noise. In this study, the performance of $TE^{\theta }$ estimators reduce with an increased in noise power, especially for $TE_{KSG}^{\theta }$ and $TE_{k\alpha }^{\theta }$, which cannot detect the coupling with a high noise power. This may be due to the discretization in $TE_{bin}^{\theta }$ and $TE_{sym}^{\theta }$, making them less sensitive to noise. For low, realistic noise, $TE_{bin}^{\theta }$, $TE_{sym}^{\theta }$, and $TE_{k\alpha }^{\theta }$ can accurately detect the directional interaction. Because of volume conduction in the brain, there may be linear mixing in adjacent brain areas. Linear mixing leads to a decrease in $dTE^{\theta }$ values for $\omega^{X\to Y}$ > 0, which could result in low sensitivity, and inflates the $dTE^{\theta }$ values, increasing the risk of false positives. These may result from the reduction of useful information in X to predict Y for $\omega^{X\to Y}$ > 0 and self-prediction information in Y added to X for $\omega^{X\to Y}$ = 0. The false positive rate of $TE_{sym}^{\theta }$ is increased with an increase in the mixing strength, which is significantly higher than that of the other methods. This may be related to the symbolization in $TE_{sym}^{\theta }$, which makes $TE_{sym}^{\theta }$ more sensitive to the change in signal asymmetry. In $TE_{sym}^{\theta }$, increasing the samples also improves the false positive rate, even for weak linear mixing. Therefore, we should pay more attention to the erroneous judgement when applying $TE_{sym}^{\theta }$ to detect the interaction between adjacent brain areas. $TE_{k\alpha }^{\theta }$ provides substantial robustness to linear mixing. This phenomenon is probably due to Renyi’s entropy, instead of Shannon entropy used in $TE_{k\alpha }^{\theta }$, and a functional defined on positive definite and infinitely divisible kernel matrices is applied to approximate Renyi’s entropy, which can capture the similarity relations among signal pairs and detect slow changing features in data [43].

The sample size is also an important factor affecting the performance of the methods. Accumulating evidence shows that accurate recognition of directional interaction relies on a sufficient sample size [44]. The results in this study show that increasing the sample size effectively improves the validity of the estimators. However, expanding the sample capability also leads to a great time cost. As shown in Figure 8, the time consumption of the estimators scales up with an increase in sample size. Because of the trace operator on Gram matrix (G) to power α ($tr\left(G^{\alpha }\right)$) in $TE_{k\alpha }^{\theta }$, it poses a great challenge in terms of both storage and computing when using $TE_{k\alpha }^{\theta }$ in practice [45]. In $TE_{bin}^{\theta }$, a sample binning method rather than more complex technology is used to reconstruct the time series state-space, so, compared with other methods, the time cost of $TE_{bin}^{\theta }$ is within an acceptable range. 

The performance baseline of the competing methods is investigated on the actual neural signals, which are recorded from PFC and V1 for a ferret in an awake state. $TE_{bin}^{\theta }$ and $TE_{sym}^{\theta }$ with a large sample size identify the bidirectional interactions between PFC and V1 and, for $TE_{bin}^{\theta }$, the interaction lag values are clustered around 5 and 20 ms for FF and around 1 ms for FB, consistent with the results in [40]. We also detect FF in the low frequency band (0–20 Hz) and FB in the high frequency band (40–60 Hz). Unexpectedly, these results are different from those in humans and monkeys. Studies have shown that a low frequency mediates FB, while a high frequency mediates FF, when humans and monkeys perform cognitive tasks such as memory encoding and recall [46,47,48]. This is consistent with the functional interpretation under the hierarchical predictive coding framework. According to predictive coding, the main function of FB is to provide predictions of input signals by integrating memory and expectations, which mainly operate on slow time scales. Meanwhile, FB operates on faster time scales, because FB has to respond to fast sensory inputs [49]. However, in the ferret experiment, although the animals receive the visual stimuli, they do not need to respond to the visual stimuli. Therefore, the inter-regional interactions of ferrets may be different from those of animals with task requirements. Another reason for the difference in results between ferrets and other animals may be that only data from one ferret are used in this manuscript, which is not statistically significant. In subsequent studies, we hope to use more data to reveal the neural mechanisms behind FF and FB of ferrets.

Accumulating evidence shows that the θ frequency mediates the spatial information processing and communication between Hp and PFC [7,50,51,52,53]. Studies investigating the interaction between brain areas of rodents suggest that the θ band mediates the spatially related information from Hp to PFC in odor–spatial associative tasks [54]. This pattern of the θ band modulation spatial information has also been reported in Hp and NCL of pigeons, supporting the formation of a stable route for homing [55]. In our report, an information flow preferentially from Hp to NCL at the θ band is detected by $TE_{bin}^{\theta }$, indicating that, like in mammals, the spatially related information is transferred from Hp to NCL for decision-making, and avian Hp-NCL may have the same information processing and interaction mode as those of mammalian Hp-PFC in sensory–spatial associative learning. 

A limitation of this paper is that $TE^{\theta }$ is not suitable for neural activity with the learning process [28]. However, pigeons take a long time to learn the visual–spatial associative task (about 30 days). So, we assume that the neural activity in a session is stable. We look forward to further research on the application of $TE^{\theta }$ in slow learning processes. Just one session with a 75% to 80% accuracy rate is applied to detect the interaction between Hp and NCL of pigeons in visual–spatial associative learning. In the future, we will investigate the dynamics of Hp and NCL interactions in associative learning. 

## 5. Conclusions

In the study, we investigate the performance of four commonly used $TE^{\theta }$ methods ($TE_{bin}^{\theta }$, $TE_{KSG}^{\theta }$, $TE_{sym}^{\theta }$, and $TE_{k\alpha }^{\theta }$) on the simulated signal pairs and the actual neural data with known interaction properties. The results show a performance baseline table that contains the most suitable method for different scenarios. Then, $TE_{bin}^{\theta }$ is applied for the local field potential recorded from pigeons and detects an information flow predominantly from Hp to NCL at the θ band during the decision-making period in visual–spatial associative learning. These outcomes highlight the importance of choosing the appropriate method for detecting the interactions between brain regions and provide a reference for researchers when using these methods. 

## Figures and Tables

**Figure 1 entropy-25-00994-f001:**
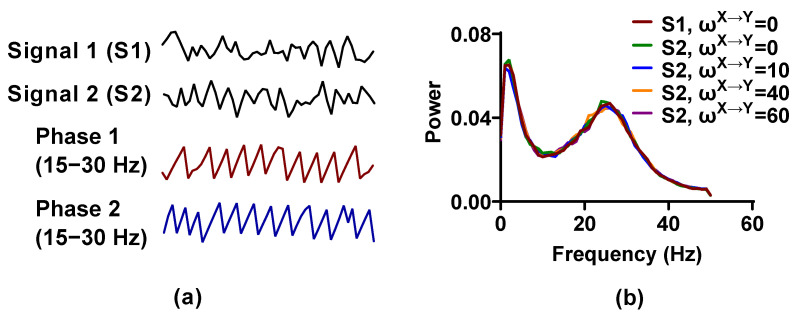
The simulated signal pairs generated by a neural mass model (NMM). (**a**) We use the NMM to produce 1 s time series of Signal 1 and Signal 2 and the instantaneous phases from 15 to 30 Hz are extracted by Hilbert transform; (**b**) power averaged over 100 simulations for each weight factor is plotted as a function of frequency.

**Figure 2 entropy-25-00994-f002:**
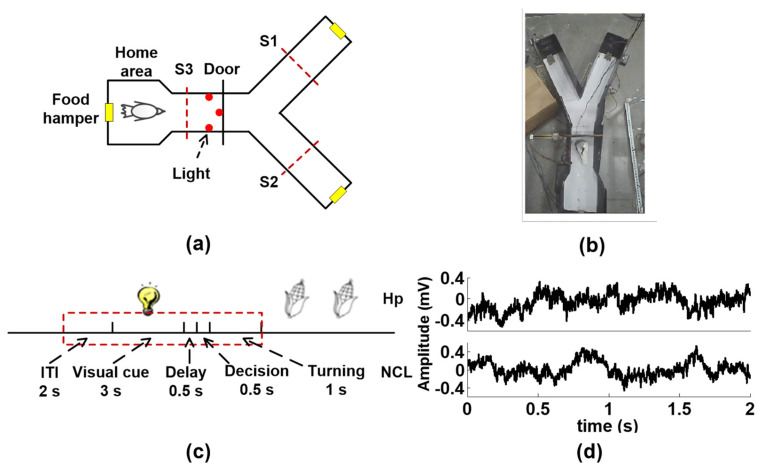
Behavioral task and experiment procedures. (**a**) Schematic diagram of the Y maze. Pigeons learn to start from the home area, then decide on the left arm or right arm according to the light color. A red light corresponds to the right arm and green light to left arm. S1, S2, and S3 are the infrared sensors. (**b**) A pigeon learning the associative task in the maze. (**c**) Diagram of the visual–spatial learning task. The red dashed line is the epochs to be analyzed. Because the animals enter into a critical decision-making place when the door is opened, 0.5 s after the end of the delay is considered to be a decision-making period. The inter trial interval (ITI) is 2 s, the visual cue period is 3 s, the delay period is 0.5 s, the decision-making period is 0.5 s, and the tuning period is 1 s. (**d**) Examples of simultaneous local field potential signals (LFPs) recorded from the hippocampus (Hp) and nidopallium caudolaterale (NCL).

**Figure 3 entropy-25-00994-f003:**
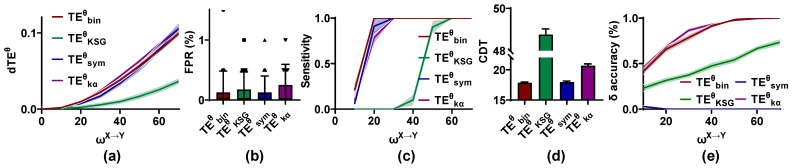
Accuracy of the phase transfer entropy ($TE^{\theta }$) methods in identifying and quantifying the directional interaction. (**a**) The differential $TE^{\theta }$ ($dTE^{\theta }$) values of these methods increase monotonically with weight factor $\omega^{X\to Y}$ from 0 to 70. (**b**) False positive rate (FPR) values of the competing methods have no significant difference and are lower than 0.01. (**c**) The sensitivity of the estimators is plotted against $\omega^{X\to Y}$, and that of $TE_{bin}^{\theta }$ is stronger than that of other methods. (**d**) The coupling detection threshold (CDT) values of $TE_{KSG}^{\theta }$ are significantly larger than those of other methods. (**e**) The interaction lag (δ) accuracy for these estimators is improved with an increase in $\omega^{X\to Y}$, and $TE_{sym}^{\theta }$ cannot identify the interaction lag correctly. All results (except for the interaction lag accuracy) in Figure 3, Figure 4, Figure 5, Figure 6, Figure 7, Figure 8, Figure 9 and Figure 10 are presented as mean ± s.d. (shading or error bar) for multiple estimates.

**Figure 4 entropy-25-00994-f004:**
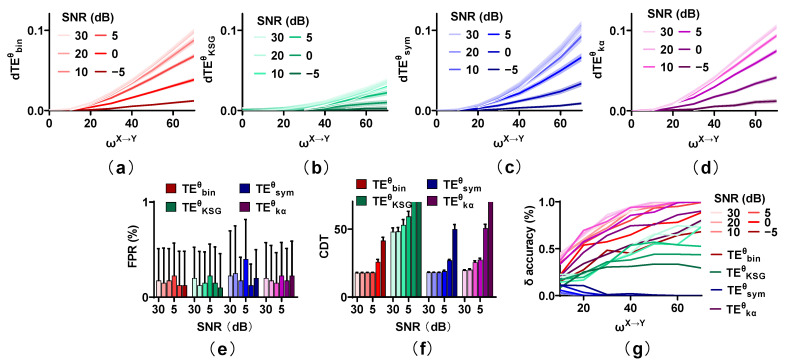
Robustness of the $TE^{\theta }$ methods on noise. (**a**–**d**) Noise reduces the $dTE^{\theta }$ values of estimators, especially for the strong noise. (**e**) The FPR values of these methods remain at a low level regardless of the noise power. (**f**) The CDT values of the four methods are increased with the SNR changing from 30 to −5 dB. (**g**) The δ accuracy of these estimators shows the same trend as that of the sensitivity and reduces with an increase in noise power.

**Figure 5 entropy-25-00994-f005:**
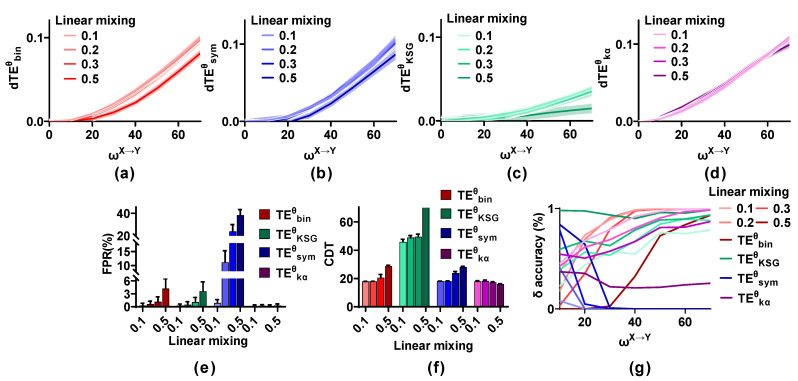
Stability of the $TE^{\theta }$ methods in detecting the directional interaction in the presence of linear mixing. (**a**–**d**) Strong linear mixing (m = 0.5) reduces the $dTE^{\theta }$ metric values of the estimators, except for $TE_{k\alpha }^{\theta }$. (**e**) The FPR values of $TE_{bin}^{\theta }$, $TE_{KSG}^{\theta }$, and $TE_{sym}^{\theta }$ are affected by strong linear mixing, but those of $TE_{k\alpha }^{\theta }$ remain at a low level. (**f**) The CDT values of the $TE^{\theta }$ methods increase with m from 0.1 to 0.5, but those of $TE_{k\alpha }^{\theta }$ are stable regardless of m. (**g**) Linear mixing affects the δ accuracy of the estimators.

**Figure 6 entropy-25-00994-f006:**
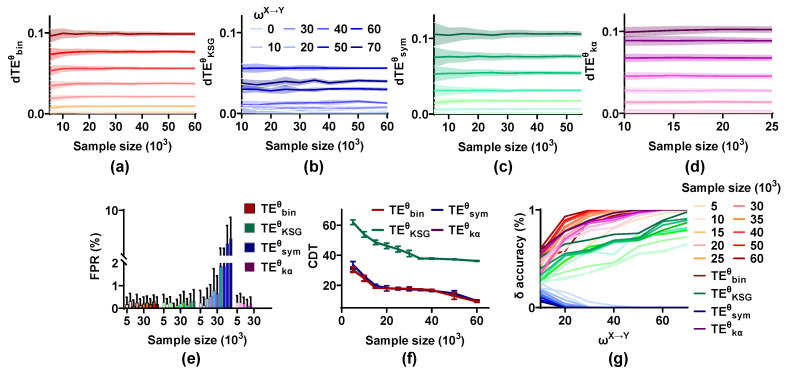
The sample size improves the effectiveness of the $TE^{\theta }$ methods. (**a**–**d**) The mean values of $dTE^{\theta }$ for all estimators remain stable, but the variances reduce with an increase in sample size. (**e**) The FPR values of $TE_{bin}^{\theta }$, $TE_{KSG}^{\theta }$, and $TE_{k\alpha }^{\theta }$ fluctuate around 0.01 for varied sample size, while those of $TE_{sym}^{\theta }$ are increased from 0.01 to 0.05. (**f**) The sample size has a positive effect on the CDT values of methods. (**g**) The δ accuracy of these methods improves with an increase in sample size, except for that of $TE_{sym}^{\theta }$.

**Figure 7 entropy-25-00994-f007:**
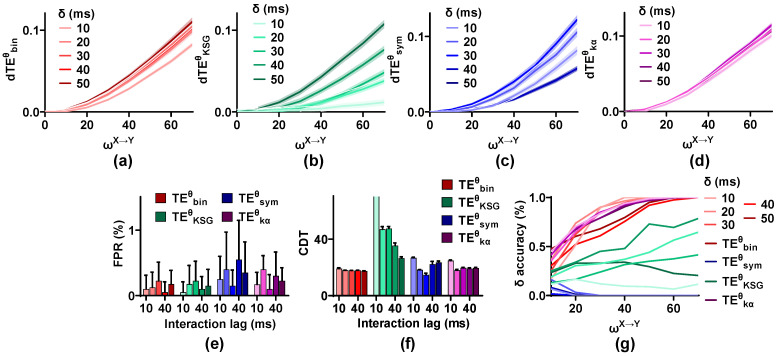
Robustness of the $TE^{\theta }$ methods on interaction lag. (**a**–**d**) The $dTE^{\theta }$ values of these estimators show different changing characteristics for varied interaction lag. (**e**) The FPR values of the $TE^{\theta }$ methods are below 0.01 for any δ value. (**f**) The CDT values of the $TE^{\theta }$ methods are affected by interaction lag. (**g**) The δ  accuracy of these estimators has complex characteristics for varied interaction lag.

**Figure 8 entropy-25-00994-f008:**
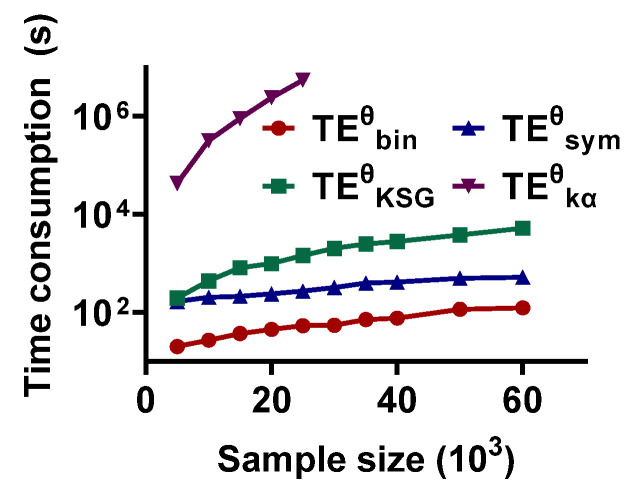
The computational complexity of the $TE^{\theta }$ methods. They are calculated for a complete estimation, which includes scanning the interaction lag in the range of 10 to 70 ms, estimating the $dTE^{\theta }$ value, and constructing the null hypothesis distribution.

**Figure 9 entropy-25-00994-f009:**
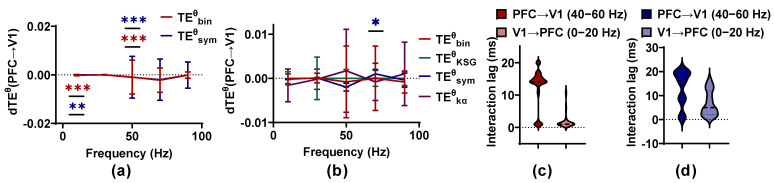
Performance baseline of the $TE^{\theta }$ methods on the actual neural data with known interaction properties. (**a**,**b**) The $dTE^{\theta }$ values of the $TE^{\theta }$ estimators are calculated for varied frequency bands with a large (**a**) and small sample size (**b**), respectively. ‘*’ represents a significant interaction between PFC and V1. ‘*’ above curves indicates the information flow preferentially from PFC to V1, and ‘*’ below curves indicates the interaction from V1 to PFC. ‘***’ p < 0.001, ‘**’ p < 0.01, ‘*’ p < 0.05. (**c**) The δ values of $TE_{bin}^{\theta }$ with a large sample size are clustered around 5 and 20 ms for PFC → V1 and around 1 ms for V1 → PFC. (**d**) The δ values of $TE_{sym}^{\theta }$ with a large sample size are more dispersed than those of $TE_{bin}^{\theta }$.

**Figure 10 entropy-25-00994-f010:**
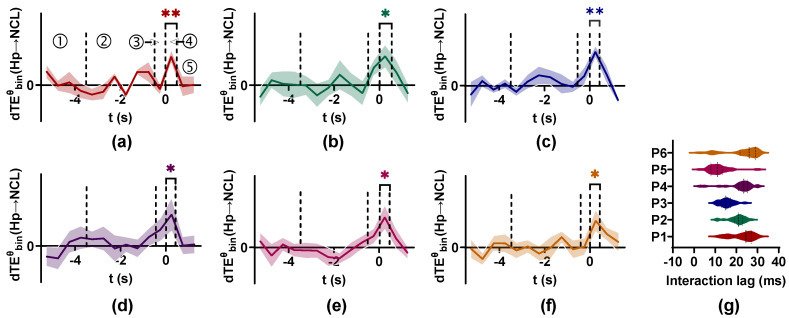
The interaction between Hp and NCL of pigeons in visual–spatial associative learning. (**a**–**f**) The $dTE^{\theta }$ values from Hp to NCL for six pigeons (p21, p22, p23, p24, p25, and p26) are calculated by $TE_{bin}^{\theta }$. The information flow is predominantly from Hp to NCL at the θ band in the decision-making period. The solid line is the mean value of $dTE^{\theta }$ and the shaded area represents the standard deviation of $dTE^{\theta }$, ‘**’ p < 0.01, ‘*’ p < 0.05. ①, ②, ③, ④, and ⑤ indicate the ITI period (2 s), the visual cue (3 s), the delay period (0.5 s), the decision-making period (0.5 s), and the turning period (1 s), respectively. Time 0 means the end of the delay period and the point at which the animal enters the critical decision-making place. (**g**) The interaction lag between Hp and NCL of pigeons in visual–spatial associative learning.

**Table 1 entropy-25-00994-t001:** Detailed information on pigeons, trials, correct rate, and available channels.

Pigeon Number	Total Trials	Correct Trial	Correct Rate	Available Channels (Hp/NCL)
p21	33	26	78.8%	16/16
p22	35	28	80%	16/16
p23	42	32	76.2%	16/16
p24	37	29	78.4%	15/16
p25	46	35	76.1%	15/16
p26	30	23	76.7%	15/16

**Table 2 entropy-25-00994-t002:** Performance baseline table for the $TE^{\theta }$ methods. ‘
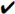
’ means the FPR is below 0.01 and CDT values are lower than 25. ‘
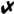
’ represents the FPR in the range of 0.01 to 0.05 or CDT values from 25 to 40. ‘
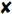
’ indicates the FPR is higher than 0.05 or CDT values are larger than 40.

	TEbinθ	TEKSGθ	TEsymθ	TEkαθ
Accuracy on the directional interaction	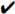	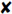	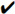	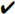
Accuracy on the interaction lag	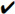	** 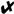 **	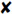	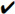
Robustness to noise	SNR ≥ 5 dB	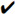	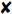	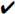	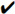
SNR < 5 dB	** 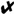 **	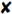	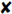	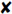
Robustness to linear mixing	m ≤ 0.3	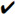	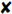	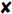	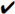
m > 0.3	** 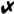 **	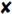	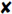	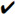
Stability to the sample size	Sample size ≤ 10 × 10^3^	** 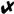 **	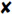	** 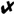 **	** 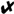 **
Sample size > 10 × 10^3^	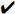	** 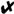 **	** 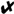 **	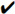
Stability to the interaction lag	Short interaction lag	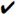	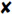	** 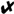 **	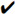
Long interaction lag	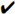	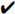	** 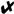 **	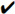
Computational complexity	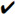	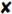	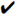	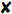

**Suitable for the following:** (1) $TE_{bin}^{\theta }$—strong noise, large sample size, any interaction lag; (2) $TE_{KSG}^{\theta }$—small sample size, long interaction lag; (3) $TE_{sym}^{\theta }$—weak linear mixing, large sample size; (4) $TE_{k\alpha }^{\theta }$—strong linear mixing, small sample size, any interaction lag.

## Data Availability

The local field potential signals recorded from pigeons are available upon request from the corresponding author.

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
