# Peer review of "Performance Baseline of Phase Transfer Entropy Methods for Detecting Animal Brain Area Interactions"

_entropy, 2023, doi:10.3390/e25070994_

Round 1

Reviewer 1 Report

In this paper, the authors summarize several phase transfer entropy methods and validate their use using several simulation examples including neural mass models and also field potential recordings from ferrets and pigeons. The methods are well described, simulations are detailed, and LFP recordings also corroborate the simulation findings. I have some comments:

(i) In Figures 9c and 9d, the authors compare the feedforward flow in one frequency (low frequency) to feedback flow in another frequency (gamma frequency). This is problematic due to the 1/f scaling effect of transfer entropy, which becomes difficult to estimate as frequency increases. Therefore, comparison of results should be shown for both feedforward and feedback transfer entropy in each frequency separately.

(ii) The low frequency feedforward results in Figures 9 and 10 is not sufficiently motivated. Also, the implications of this feedforward and feedback information flow is not discussed. The authors should discuss their results in the context of the following references which discuss low frequency feedforward and higher frequency feedback information flow in the brain.

[1]https://www.sciencedirect.com/science/article/pii/S089662731401099X

[2]https://www.sciencedirect.com/science/article/pii/S0896627315011204

[3]https://www.jneurosci.org/content/41/40/8427.long

[4]https://academic.oup.com/cercor/article/32/23/5343/6524032

[5]https://www.pnas.org/doi/10.1073/pnas.1515657113

(iii) The classic Schreiber reference on transfer entropy should also be included:

https://journals.aps.org/prl/abstract/10.1103/PhysRevLett.85.461 

Reviewer 2 Report

Upon reviewing the article, it appears to lack a clear and concise research question, making it difficult for the reader to understand the purpose of the study. Additionally, there seem to be inconsistencies and gaps in the results section, calling into question the validity and reliability of the findings.

Furthermore, the article could benefit from more detailed and rigorous analyses, as well as a thorough discussion of the implications of the findings for the field. The literature review also requires a more comprehensive approach, with a broader range of studies and theories cited to support the research hypothesis.

Finally, the article would benefit from a more engaging writing style, including clearer structure, more concise language, and a greater use of visuals and examples to illustrate key points. Overall, there is room for improvement in terms of the coherence, accuracy, and impact of the research presented.

Reviewer 3 Report

Manuscript ID: entropy-2391010

Title: Baseline of phase transfer entropy methods for detecting animal brain area interactions

Authors: Junyao Zhu, Mengmeng Li, Zhiheng Zhang, Gang Liu, and Hong Wan

Overview

This study by Zhu et al. characterized four methods for estimating the phase transfer entropy using simulated data from a neural mass model (NMM) and in vivo local field potential (LFP) data from awake ferrets. Using four methods, namely, Binning, KSG, Symbolic, and Renyi’s alpha-entropy estimators, authors establish the criteria for optimal implementation of aforementioned estimators on in vivo LFP data from behaving pigeons, to obtain insights about the coupling and direction of information transfer across brain regions of interest. Overall, authors have conducted well-defined analyses and tried to provide a clear picture of their results in the manuscript, although, in my opinion, there are some improvements needed to improve the manuscript as described below.

General comments (In the order of appearance)

Title

  • For the most part the meaning of the word “Baseline” is not clear and it becomes apparent after reading through the manuscript that authors are characterizing the estimation methods for phase transfer entropy. 

  • I recommend that the title should be changed to, “Characterization of phase transfer entropy methods for detecting animal brain area interactions”. Also, I suggest replacing the word “baseline" in some other places in the manuscript with other suitable words as suggested below for each section.

Abstract

  • Line 14, 18, 20, 23 - Replace word ‘baseline’ with ‘characterization’.

  • Line 15 - Replace ‘signal pairs’ with ‘data’

  • Line 16 - ‘It's not clear if ‘their’ refers to the estimation methods or data. Authors should rephrase the sentence as “….to investigate the accuracy, stability, and computational complexity of the phase transfer entropy (or TE-theta) methods in identifying…….”

Introduction

  • Line 30 - Replace ‘support’ with ‘supports’

  • Line 32 - Replace ‘homoly’ with ‘homologous’

  • Line 40 - Authors should ensure to discuss in detail how this study would help in the development of AI, in the discussion section. Add references as needed.

  • Line 42 - Replace ‘calculate’ with ‘calculating’

  • Line 43 - Replace ‘so that’ with ‘since’

  • Line 47-48 - Add refs

  • Line 49 - Authors state, “... most critical assignment is to estimate TE-theta metric value.” Describe what is the ‘metric value’. Why is its estimation critical? Please clarify. Add references as needed.

  • Line 51 - Define theta-x, theta-y for equations, define/expand the probability terms in simple words.

  • Line 54 - Define “strips”.  (Alternatively, refer readers to line 147.)

  • Line 54 - Replace ‘neatest’ with ‘nearest’

  • Line 62 - add reference for Renyi’s alpha entropy

  • Line 68 - Replace ‘functional’ with ‘function’

  • Line 71 - Replace word ‘baseline of’ with ‘criteria for applying’.

  • Line 72 - Replace word ‘baseline’ with ‘characteristic’.

  • Line 76 - Replace word ‘baseline’ with ‘applicability’

  • Line 81 - Replace word ‘baseline’ with ‘properties’. Rephrase the sentence - “The properties of TE-theta methods for suitable application are not yet well understood.”

  • Clarify why authors choose ferret data for characterizing the methods instead of rat/monkey data? 

  • Line 85 - Remove ‘to’

  • Line 92 - Replace word ‘baseline’ with ‘applicability’

  • Line 94 - Replace word ‘baseline’ with ‘characterization’

  • Line 97 - Replace ‘signal pairs’ with ‘data’

  • Line 98 - Replace word ‘baseline’ with ‘properties’

Methods

  • Line 112 - Define ‘R’ superscript Dxd. Clarify to the readers why x ranges from “(t-u) to dx”, whereas. Y ranges from “(t-1) to dy” by mentioning a clear connection across the interaction lag (u) between X and Y time series.

  • Line 113 - Deine ‘N’. 

  • Line 113 - Define ‘embedding’ for d and tau.

  • Line 153 - Explain/clarify why k is generally 4? Provide reference as needed.

  • Line 167 - Explain/clarify why the embedding dimension ‘d’ is set to 3 and the embedding delay ‘tau’ is set to 12? Provide reference as needed.

  • Line 172 - For the first term of eq11, define the term ‘tr’  in the denominator.

  • Line 179-180 - Explain/clarify why d=2 and tau=1 are used to reconstruct space states. Why alpha =3 for Renyi’s alpha-entropy?

  • Line 202 - Replace ‘interesting’ with ‘interest’

  • Line 208 - Is there a mathematical basis for prominence of beta frequency in the data generated by NMMs? 

  • Line 217, 226, and 250 - The simulated data from NMM is 1s long to obtain Hilbert transform. To evaluate the stability of TE-theta methods in presence of noise and linear mixing, signal pairs are 2s long. Ferret LFP data is separated into 4.8 s long epochs. The in vivo pigeon data is 8s long. Why is there a variability in the length of simulated and neural data while characterizing the TE-theta methods? Could this variability affect the results and bias the characterization of any of the methods to estimate TE-theta? Authors may discuss this in the discussion section if it seems a valid concern or clarify why this is not a significant concern.

  • Line 227 - Replace word ‘baseline’ with ‘properties’

  • Line 245 - Ferret data has no task demand - so the temporal properties of the ferret data can be different from the data for pigeons with a task demand - Could the phase transfer entropy calculations be affected due to this difference?

  • Line 246 - Replace ‘recorded’ with ‘record’. 

  • Line 246 - Replace ‘mental’ with ‘metal’. 

  • Line 252-253 - Why are the epochs with large power in the delta band rejected?

  • Line 259 - Replace word ‘baseline’ with ‘characterization’

  • Line 260 - Why 2 and 30 epochs are chosen? How different are the metric values for 2 and 30 epochs?

  • Line 261 - Why interaction lag is chosen between 1-20ms?

  • Line 273 - Replace ‘that’ with ‘when’

  • Line 274 - Specify how many trials were removed based on motion artifacts for the data used for final results. If needed, provide a table.

  • Line 284 - References for this statement, “All trials in a session are assumed to have the equivalent brain activity” - is it as well established for Pigeon brain as for mammalian models? Discuss

  • Line 285-286 - What is the direction of information flow and coupling during error trials? Error trial data could provide useful insights about the NCL-Hp coupling.

  • Figure 2d - add scale for Hp and NCL LFP traces.

Results

  • Line 303 - Replace word ‘baseline’ with ‘characterization based’. Rephrase the subtitle as, “3.1 The characterization based on the simulated data.”

  • Line 307 - Same as aforementioned comment - Why is the trial duration 2s for simulated data and not 4.8 s (Ferret data) or 8 s (Pigeon data)?

  • Figure 3, 4, 5, 6, 7, 9 - Specify what error bars represent in these figures?

  • Figure 3 - expand the meaning of omega (weight) superscript X to Y in legend.

  • Line 328-329 - Authors may clarify/discuss the reasons for observed differences across the four methods for estimating TE-theta.

  • Line 381-382 - Rephrase as, “.....cannot detect the interaction lag irrespective of the effect of linear mixing (m).”

  • Line 405 - replace ‘baseline’ with ‘characterization’

  • Line 412-413 - Rephrase as, “The FPR values of …remain at a low level irrespective of the sample size.”

  • Line 432 - Replace “all” with ‘are’

  • Line 443 - 444 - Rephrase the sentence as, “Generally, the surrogate signal pairs are shuffled 200 times to construct a null hypothesis distribution.”

  • Line 467 - replace ‘baseline’ with ‘characterization’ - Rephrase the subtitle as, “3.2 The characterization of TE-theta methods based on the neural data”

  • Line 467 - replace ‘baseline’ with ‘properties’

  • Line 491 - replace ‘baseline’ with ‘applicability criteria’

  • Figure 9 - Line 493 - replace ‘Baseline’ with ‘Characterization’

  • Line 501 - Line 467 - Remove ‘baseline’. Rephrase subtitle as, “Implementing suitable phase transfer entropy estimation method on the interaction between Hp and NCL of pigeons in spatial associative learning”.

  • Line 503 - Replace ‘baseline’ with ‘characterization’

  • Line 519 - Replace ‘Baseline’ with ‘Characterization’

  • Line 533 - Replace ‘baseline’ with ‘characterization’

  • Line 547 - Replace ‘baseline’ with ‘characterization’

  • Line 547 - Replace ‘baseline’ with ‘properties’

  • Line 594 - Replace ‘baseline’ with ‘properties’ ….are investigated

  • Line 620 - Replace ‘baseline’ with ‘properties’

  • Line 622 - Replace ‘baseline’ with ‘characterization’

Improvements suggested in comments.

Round 2

Reviewer 1 Report

I don't have any further technical comments.

The English is very awkward throughout the manuscript. Please take help of a English professional to improve the English presentation.